# Discriminative Measurement of Absorbed Dose Rates in Air from Natural and Artificial Radionuclides in Namie Town, Fukushima Prefecture

**DOI:** 10.3390/ijerph18030978

**Published:** 2021-01-22

**Authors:** Koya Ogura, Masahiro Hosoda, Yuki Tamakuma, Takahito Suzuki, Ryohei Yamada, Ryoju Negami, Takakiyo Tsujiguchi, Masaru Yamaguchi, Yoshitaka Shiroma, Kazuki Iwaoka, Naofumi Akata, Mayumi Shimizu, Ikuo Kashiwakura, Shinji Tokonami

**Affiliations:** 1Graduate School of Health Sciences, Hirosaki University, 66-1 Hon-cho, Hirosaki, Aomori 036-8564, Japan; kogura@hirosaki-u.ac.jp (K.O.); m_hosoda@hirosaki-u.ac.jp (M.H.); tamakuma@hirosaki-u.ac.jp (Y.T.); suzuki-takahito@fujielectric.com (T.S.); yamada.ryohei@jaea.go.jp (R.Y.); h20gg204@hirosaki-u.ac.jp (R.N.); r.tsuji@hirosaki-u.ac.jp (T.T.); masarun@hirosaki-u.ac.jp (M.Y.); ikashi@hirosaki-u.ac.jp (I.K.); 2Institute of Radiation Emergency Medicine, Hirosaki University, 66-1 Hon-cho, Hirosaki, Aomori 036-8564, Japan; akata@hirosaki-u.ac.jp (N.A.); mshimizu@hirosaki-u.ac.jp (M.S.); 3Faculty of Education, University of the Ryukyus, 1 Senbaru, Nishihara-cho, Okinawa 903-0213, Japan; y_shiro@cs.u-ryukyu.ac.jp; 4National Institutes for Quantum and Radiological Science and Technology, 4-9-1 Anagawa, Inage, Chiba 263-0024, Japan; iwaoka.kazuki@qst.go.jp

**Keywords:** Fukushima Daiichi Nuclear Power Plant, Namie Town, natural radionuclides, artificial radionuclides, cesium-134, cesium-137, external exposure dose evaluation

## Abstract

Ten years have elapsed since the accident at the Fukushima Daiichi Nuclear Power Plant in 2011, and the relative contribution of natural radiation is increasing in Fukushima Prefecture due to the reduced dose of artificial radiation. In order to accurately determine the effective dose of exposure to artificial radiation, it is necessary to evaluate the effective dose of natural as well as artificial components. In this study, we measured the gamma-ray pulse-height distribution over the accessible area of Namie Town, Fukushima Prefecture, and evaluated the annual effective dose of external exposure by distinguishing between natural and artificial radionuclides. The estimated median (range) of absorbed dose rates in air from artificial radionuclides as of 1 April 2020, is 133 (67–511) nGy h^−1^ in the evacuation order cancellation zone, and 1306 (892–2081) nGy h^−1^ in the difficult-to-return zone. The median annual effective doses of external exposures from natural and artificial radionuclides were found to be 0.19 and 0.40 mSv in the evacuation order cancellation zone, and 0.25 and 3.9 mSv in the difficult-to-return zone. The latest annual effective dose of external exposure discriminated into natural and artificial radionuclides is expected to be utilized for radiation risk communication.

## 1. Introduction

On 11 March 2011, a magnitude 9.0 earthquake struck the Tohoku region along the eastern coast of Japan. The earthquake caused a tsunami with a height of more than 15 m, and affected the Fukushima Daiichi Nuclear Power Plant (FDNPP). The FDNPP lost power and the cores of Units 1 to 3 became heated and melted. This caused a hydrogen gas explosion [1]. As a result of the FDNPP accident, ^132^Te, ^131^I, ^134^Cs, ^137^Cs, and rare gases such as ^133^Xe, etc., were released into Fukushima Prefecture and other eastern regions of Japan [2]. The radioactivity of radionuclides released into the atmosphere is shown in the UNSCEAR 2013 report (Table 1) [3]. On the day of the accident, the Japanese government issued an indoor evacuation order to residents within 10 km of the FDNPP, and issued an evacuation order to residents within 20 km the next day [4]. Thereafter, the area where the annual cumulative dose may have exceeded 20 mSv, outside the 20 km area from the FDNPP was designated as a “planned evacuation zone”. In addition, regardless of the annual cumulative dose, the area within 20 to 30 km of the FDNPP was designated as an “emergency evacuation preparation zone” and the area within 20 km was designated as a “warning zone” [5]. Namie Town, Fukushima Prefecture (The location map that is shown in Figure 1a was made by original maps from d-maps.com), is also one of the areas significantly contaminated by radionuclides due to the FDNPP accident, and because it was a planned evacuation zone, the townspeople living there were forced to evacuate. In 2012, the area where the annual cumulative dose was confirmed to be 20 mSv or less was designated as an “evacuation order cancellation preparation zone”. This is the area where temporary return homes, restricted businesses such as shops, hospitals, and farming are permitted. Areas where the annual cumulative dose may exceed 20 mSv but are confirmed to be 50 mSv or less have been designated as a “restricted residence zone” and it has become possible to temporarily return home or enter for road restoration. Areas where the annual cumulative dose exceeds 50 mSv and the annual cumulative dose may not fall below 20 mSv, five years from 2012, has been designated as a “difficult-to-return zone”. Figure 1b indicates each area division, and taken from the official website of Fukushima Prefecture [5]. Subsequently, the artificial decontamination of radionuclides was actively promoted, and in 2017, six years after the earthquake, evacuation orders were lifted in some areas of Namie Town [6]. Currently, the return of evacuees is progressing, and by the end of November 2020, more than 1500 people were living in Namie Town [7]. Before the Great East Japan Earthquake, the registered population of Namie Town was 21,434 [8]. Years after the FDNPP accident, the returning residents continue to have a significant amount of radiation anxiety [9]. Experts in radiation science and psychology at each Japanese support organization, including the university of the current authors, have communicated radiation risk, and interacted with residents to reduce anxiety about radiation. In consideration of this, Kudo et al. conducted a questionnaire survey on the basic knowledge of radiation among those who returned to Namie Town. It was found that many Namie townspeople recognize that natural and artificial radiation have different effects on the human body, even if the effective dose is the same [10].

Since the FDNPP accident, national staff and researchers at universities and research institutions have been evaluating artificial radioactive contamination and investigating the distribution of ambient dose equivalent rates [11,12,13]. In addition, internal and external exposures from artificial radionuclides are being evaluated [14,15,16,17,18,19], and monitoring posts are installed in various locations to continuously measure the ambient dose equivalent rate [20]. In 2017, Shiroma et al. conducted a car-borne survey in Namie Town, Fukushima Prefecture, and reported that the absorbed dose rate in air was 0.041–11 µGy h^−1^ [21]. More than nine years have passed since the FDNPP accident, and the relative contribution of natural radiation to ambient dose equivalent rates is increasing because the dose of artificial radiation is decreasing. This means that it is not possible to estimate the effects on the human body due to artificial radionuclides, without correctly evaluating the dose from natural radionuclides. People with a high risk of internal exposure, such as agricultural workers, need information on internal exposure due to inhalation of dust. However, clarifying the actual conditions of external exposure from natural and artificial radionuclides is useful for radiation risk communication for general population, which has a low risk of internal exposure. In this study, the gamma-ray pulse-height distribution was measured and analyzed in Namie Town, which was divided into 1 km × 1 km meshes. An absorbed dose rate map that discriminated between natural and artificial radionuclides was created from the absorbed dose rate in the air, and the annual effective dose to external exposure was calculated.

## 2. Materials and Methods

### 2.1. Measurement Location and Method of γ-Ray Pulse-Height Distribution

From 15 September 2016 to 13 December 2019, gamma-ray pulse-height distributions were obtained at the 130 accessible points that divided the entire area of Namie Town into a mesh of 1 km × 1 km. A 3 × 3-inch NaI(Tl) scintillation spectrometer (EMF-211, EMF Japan Co., Himeji, Japan [22]) was used to obtain the measurements. The detector was installed 1 m above the ground and connected to a control laptop PC. The measurement time was 900 s. Latitude and longitude coordinate data were obtained using a Global Positioning System to create an absorbed dose rate map. Gamma-ray pulse-height distributions at 2–5 points were additionally acquired in six of the 130 meshes, and the fluctuation of the absorbed dose rate in air in the mesh was evaluated.

### 2.2. Analysis of Gamma-Ray Pulse-Height Distribution and Correction of Absorbed Dose Rate in Air

The gamma-ray pulse-height distributions obtained by the NaI(Tl) scintillation spectrometer is different from the distributions of the gamma-ray energy spectrum. The pulse-height distributions of gamma-ray are unfolded into the energy spectrum by a response matrix of 49 rows × 49 columns, and then the dose contributions for each radionuclide are calculated according to the previous reports to discriminate between natural and artificial radionuclides [23,24,25]. The absorbed dose rate in air obtained by the analysis needs to be corrected to consider the number of days elapsed from the measured date. Factors that reduce radioactivity in the environment include the physical half-life of radionuclides, diffusion by wind, rain, and infiltration into soil, and the implementation of artificial decontamination of radioactive substances. In order to comprehensively evaluate the factors that affect the attenuation of radioactivity, the apparent half-life was calculated using the data of the air dose rate that is regularly observed at the monitoring posts widely installed in Namie Town. There are 103 monitoring posts in Namie Town, and the measurement data are published on the website [20]. Some of these datasets have long-term data loss within the period in which we measured the gamma-ray pulse-height distribution, and significant dose increases and decreases in a short period of time that are not due to artificial decontamination. It is probable that the data loss could not be measured due to maintenance of the monitoring posts. The short-term significant fluctuation of the ambient dose equivalent rate may be due to a device malfunction, but the specific cause is unknown. These data may affect the appropriate time decay correction of absorbed dose rates in air. Therefore, the apparent half-life was calculated using the data of 55 monitoring posts, and excluding the lossy dataset and coefficient of determination R^2^ of less than 0.7 (not due to artificial decontamination) in the exponential approximation of the ambient dose equivalent rate. Equation (1) was used to calculate the apparent half-life (*T_a_*).
(1)$$T_{a}=t\;\times \;\frac{0.693}{ln\left(\frac{D_{1}}{D_{0}}\right)}$$
where *D*_0_ and *D*_1_ are the ambient dose equivalent rates (µSv h^−1^) as of 1 April 2016, and 1 April 2020, respectively, and t is the elapsed time, which was taken as used four years. The FDNPP accident released short half-life radionuclides such as ^131^I and ^133^Xe and long half-life radionuclides such as ^134^Cs and ^137^Cs. Originally, it was necessary to calculate the apparent half-life for each of the short-half-life and long-half-life radionuclides, but now that nine years have elapsed since the accident, the contribution from the short-half-life radionuclides can be ignored [26,27]. The apparent half-life was calculated using the simple formula in Equation (1), considering only the contribution from radionuclides with a long half-life. The calculated apparent half-life was divided into an evacuation order cancellation zone and a difficult-to-return zone, and the fluctuation was evaluated to examine the application to the correction of the absorbed dose rate in air.

### 2.3. Estimating the Effective Dose of External Exposure

The annual effective dose of external exposure in Namie Town was estimated using Equation (2), and the time-corrected absorbed dose rate in air.
*E* = *D* × *DCF* × *T* × (*Q*_*in*_ × *R* + *Q*_*out*_)(2)
where *D* is the time-corrected absorbed dose rate in air (nGy h^−1^) and *DCF* is a dose conversion factor (Sv Gy^−1^) from the absorbed dose rate in air to the effective dose to external exposure. The natural radionuclide component *DCF* uses 0.748, as reported by Moriuchi et al., and the artificial radionuclide uses 0.73, as reported by Omori et al. [28,29]. *T* is the number of hours per year, which is 8766 h (24 h × 365.25 d). *Q*_*in*_ is the indoor occupancy factor, *Q*_*out*_ is the outdoor occupancy factor, and they are 0.83 and 0.17, respectively, as reported by Ploykrathok et al. [30]. *R* is a reduction factor, the natural radionuclide is 1, and the artificial radionuclide is 0.43, as reported by Yoshida et al. [31].

## 3. Results and Discussion

### 3.1. Absorbed Dose Rate in Air and Dose Rate Map

The gamma-ray pulse-height distribution was measured over the entire accessible area of Namie Town and was developed using a response matrix to determine the absorbed dose rate in air. The absorbed dose rates in air of the natural radionuclides, artificial radionuclides, and their totals are 15–68, 14–11,861, and 47–11,900 nGy h^−1^, respectively. The total absorbed dose rate in air obtained in this study is almost in agreement with the 0.041–11 µGy h^−1^ measured by Shiroma et al. [21]. The absorbed dose rates in air of natural radionuclides, artificial radionuclides, and their totals in the evacuation order cancellation zone are 19–51, 14–2010, and 47–2040 nGy h^−1^, respectively. The natural, artificial, and their total absorbed dose rates in air in the difficult-to-return zone are 15–68, 140–11,861, and 186–11,900 nGy h^−1^, respectively. The radioactivity ratios of cesium (^134^Cs/^137^Cs) released from Units 1, 2, and 3 of the FDNPP were reported to be 0.941, 1.082, and 1.046, respectively [32]. This radioactivity ratio is evaluated as the value as of 11 March 2011. As a result of estimating ^134^Cs/^137^Cs as of March 2011 for the measured data, the median (range) was 1.07 (1.04–1.09), and it was confirmed that ^134^Cs and ^137^Cs were released from FDNPP. The apparent half-life was calculated by analyzing the datasets of 55 monitoring posts installed in Namie Townin order to time-correct the measured absorbed dose rate in air. A total of 32 of them were located in areas exceeding 1.0 µGy h^−1^ as of April 2016. 10 of them were located in areas exceeding 1.0 µGy h^−1^ as of April 2020. The mean ± standard deviation, coefficient of variation, and median (range) of apparent half-lives in the difficult-to-return zone are 4.2 ± 1.4 y, 33%, and 4.7 (4.0–4.8) y, respectively (Appendix A
Table A1). Considering that the half-life of ^137^Cs is approximately 30 years, the reason why the apparent half-life is shortened is seemingly strongly influenced by diffusion due to environmental factors. The mean ± standard deviation, coefficient of variation, and median (range) of the apparent half-life in the evacuation order cancellation zone are 4.8 ± 2.7 y, 56%, and 4.7 (2.3–6.7) y, respectively. It was found that there are variations in the areas where residence is allowed. The apparent half-life was calculated using the data from 1 April 2016 to 1 April 2020. A detailed review of the data for each monitoring post revealed that some areas were decontaminated after April 2016, and some were decontaminated prior to that date [33]. The implementation of artificial decontamination contributes to rapid dose reduction and significantly shortens the apparent half-life. Therefore, the evacuation order cancellation zone was further divided into areas where decontamination was conducted before, and on and after, April 2016, and the apparent half-life was analyzed. Figure 2 indicates the difficult-to-return zone, evacuation order cancellation zone decontaminated before April 2016, and evacuation order cancellation zone decontaminated on, and after, April 2016 areas. The mean ± standard deviation, coefficient of variation, and median (range) of the apparent half-life in the evacuation order cancellation zone are 6.4 ± 2.0 y, 31%, and 6.1 (5.0–7.5) y, respectively (Appendix A
Table A2). Conversely, the mean value ± standard deviation, coefficient of variation, and median (range) of the apparent half-life limited to the zones where decontamination was completed after 1 April 2016, are 2.0 ± 0.6 y, 30%, 1.8. (1.6–2.3) y, respectively (Appendix A
Table A3). A significant difference test was performed using the Mann–Whitney *U* test for the apparent half-life of the evacuation order cancellation zone decontaminated before, and on and after, April 2016. It was confirmed there was a significant difference between the two groups (*p*-value < 3.8 × 10^–7^). This result demonstrates that the implementation of decontamination significantly contributes to the reduction of the ambient dose equivalent rates from artificial radionuclides. In addition, it was found that the evacuation order cancellation zone can be evaluated with a fluctuation of approximately 30%, by dividing it into two areas for the calculations. This coefficient of variation is significantly lower than when the evacuation order cancellation zone was not divided into two. In addition, a significant difference in apparent half-life was determined using the Mann–Whitney *U* test for the difficult-to-return zone and evacuation order cancellation zone decontaminated before April 2016, for the difficult-to-return zone and the evacuation order cancellation zone decontaminated on and after April 2016. The *p*-values are 6.9 × 10^−4^ and 9.5 × 10^−4^, respectively, confirming that there is a significant difference in distribution. Hayes et al. reported that the effective half-life of radiocesium in the environment was 7.8 years as a theoretical value and 3.2 years as a measured value [34]. Table 2 shows a comparison of the apparent half-life calculated in this study, the previously reported effective half-life, and the theoretical half-life.

The measured data of absorbed dose rates in air from artificial radionuclides were corrected to the values as of 1 April 2020 using different apparent half-lives for each of the three areas (Appendix B). The median (range) is shown in Table 3, and the distribution of the absorbed dose rate in air of the artificial radionuclides collected as of 1 April 2020 is shown in Figure 3.

A significant difference test was performed using the Mann–Whitney *U* test on the absorbed dose rates in the air from artificial radionuclides in the evacuation order cancellation zone and the difficult-to-return zone. It was confirmed that the two groups are significantly different (*p*-value = 6.0 × 10^−14^). The evacuation order cancellation zone is an area that the Japanese government has determined people can live in because it has been confirmed that the ambient dose equivalent rate has decreased [6]. In contrast, the difficult-to-return zone is an area where the annual cumulative dose exceeds 50 mSv as of April 2012, and the annual cumulative dose may not fall below 20 mSv after five years have elapsed [5]. It was found that the absorbed dose rate in air remained high in the difficult-to-return zone nine years after the FDNPP accident. The mean ± standard deviation and median (range) of absorbed dose rates in air by natural radionuclides throughout Namie Town are 35 ± 10 and 34 (28–42) nGy h^−1^, respectively. The national average in Japan is reported to be 50 nGy h^−1^ [36]. It was found that the average value of Namie Town was 70% of the national average value. These data can be used for radiation risk communication. The absorbed dose rate maps (Figure 4a,b) were developed so that the absorbed dose rate in air could be visually understood by dividing it into natural and artificial radionuclides.

The activity concentrations of ^40^K, ^232^Th, and ^238^U are shown in Appendix B. When examining the absorbed dose rate in air from natural radionuclides (Figure 4a), it can be seen that the eastern coastal area of Namie Town is less than 40 nGy h^−1^ in most areas. The range of activity concentrations of ^40^K, ^232^Th, and ^238^U in the evacuation order cancellation zone were 109–444, 9–32, and 9–34 Bq kg^−1^, respectively. Conversely, in the mountainous areas on the west side, there are many areas of 40 nGy h^−1^ or more. The range of activity concentrations of ^40^K, ^232^Th, and ^238^U in the difficult-to-return zone were 99–1830, 9–46, and 10–161 Bq kg^−1^, respectively. On the west side of Namie Town, where granite is widely distributed, the activity concentrations of ^40^K, ^232^Th, and ^238^U tended to be high [37]. When examining the absorbed dose rate in air from artificial radionuclides (Figure 4b), it can be seen that there is a clear difference between the coastal areas on the east side and the mountainous areas on the west side. This is a clear result of the evacuation order cancellation zone and the difficult-to-return zone. In the coastal area, decontamination was actively conducted in order to realize the return of evacuees, and the evacuation order was lifted in March 2017 [6]. In contrast, the mountainous area on the west side has many areas exceeding 1.0 µGy h^−1^, and is remains designated as a difficult-to-return zone. This result indicates that artificial decontamination activities contribute significantly to dose reduction. However, there were two meshes in the evacuation order cancellation area that exceeded 1.0 µGy h^−1^. Factors that increased the absorbed dose rate in air in this area include the presence of slopes composed of soil and the presence of localized forest areas in the city, such as bamboo groves. Slopes composed of soil have not been actively decontaminated because they may loosen the ground and cause sediment-related disasters. Local forest areas in the city, such as bamboo groves, are difficult to decontaminate by removing the upper part of the soil without cutting, which is a factor that increases the absorbed dose rate in air. However, local forests and slopes composed of soil do not always exist uniformly within a 1 km × 1 km mesh. In order to examine the variation of the measurement data in the mesh, the absorbed dose rate in air was additionally measured at 2–5 points in six out of the 130 meshes (Table 4). Although there are some fluctuations depending on the mesh, it was found that it is possible to evaluate with a volatility of approximately 50% or less. It was also determined that the volatility is not dose-dependent.

### 3.2. Estimating External Exposure Dose

Table 5 indicates the median (range) of the annual effective dose of external exposure calculated from the absorbed dose rate in the air. The annual effective doses of natural radionuclides in the evacuation order cancellation zone, difficult-to-return zone, and Namie Town as a whole are 0.12–0.33, 0.10–0.45, and 0.10–0.45 mSv, respectively, and their geometric mean (mean ± standard deviation) is 0.20 (0.20 ± 0.05), 0.24 (0.24 ± 0.06), and 0.22 (0.23 ± 0.06), respectively. The national average effective annual dose of ground gamma-rays in Japan is 0.33 mSv. It was found that the average value for the town of Namie is 70% of the national average [38,39]. The annual effective doses of external exposure to artificial radionuclides in the evacuation order cancellation zone, difficult-to-return zone, and entire Namie Town are 0.03–4.6, 0.23–19.6, and 0.03–19.6 mSv, respectively. The median annual external exposure effective dose from artificial radionuclides in the evacuation order cancellation zone (0.40 mSv) is 0.21 mSv, which differs from the median natural radionuclides (0.19 mSv). In contrast, the median annual external exposure effective dose from artificial radionuclides in the difficult-to-return zone (3.9 mSv) is 15.6 times higher than the median from natural radionuclides (0.25 mSv). A significant difference test was performed using the Mann–Whitney *U* test on the annual effective dose of external exposure from artificial radionuclides in the evacuation order cancellation zone and the difficult-to-return zone. The two groups have a statistically significant difference (*p*-value < 6.0 × 10^−^^14^). This difficult-to-return zone is an area where access to people is restricted. Cars are allowed on some sections, but the general public is still not allowed to stay for a long time [40]. Currently, in difficult-to-return zone, active decontamination is being carried out so that people can live. In the future, this artificial decontamination is expected to reduce the absorbed dose rate in air.

## 4. Conclusions

The absorbed dose rate in air was measured by discriminating between natural and artificial radionuclides in the entire area of Namie Town, an area affected by the FDNPP accident. The following results were obtained from this study:From the measurements of ^134^Cs and ^137^Cs concentrations, it was confirmed that Namie Town was radioactively contaminated by artificial radionuclides from the FDNPP accident.From the data of the monitoring posts installed in Namie Town, the median (range) of the apparent half-life of artificial radionuclides in the evacuation order cancellation zone decontaminated before April 2016, the evacuation order cancellation zone decontaminated after April 2016, and the difficult-to-return zone, is 6.4 ± 2.0, 2.0 ± 0.6, and 4.2 ± 1.4 y, respectively.The median (range) of absorbed dose rates in the air from artificial radionuclides time-corrected as of 1 April 2020, using the apparent half-life are 133 (67–511) and 1306 (892–2081) nGy h^−1^ in the evacuation order cancellation zone and the difficult-to-return zone, respectively.The median annual effective doses of external exposures from natural and artificial radionuclides are 0.19 and 0.40 mSv in the evacuation order cancellation zone and 0.25 and 3.9 mSv in the difficult-to-return zone.

Examination of the absorbed dose rate in the air from artificial radionuclides revealed a clear difference between the eastern coastal area and the western mountainous area. This result suggests that artificial decontamination activities contribute significantly to dose reduction. The distribution map of the absorbed dose rate in air measured in this study, and the information on the annual external exposure effective dose calculated by discriminating between natural and artificial radionuclides, are expected to be utilized for radiation risk communication.

## Figures and Tables

**Figure 1 ijerph-18-00978-f001:**
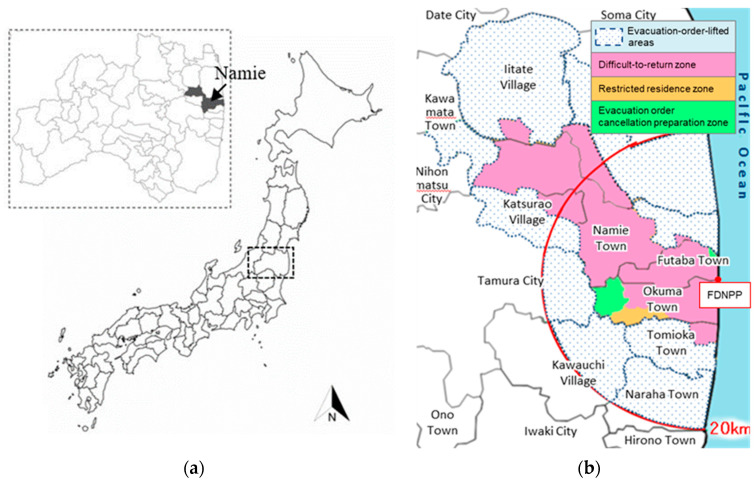
(**a**) Location of Namie Town, Fukushima Prefecture, Japan, and (**b**) officially designed evacuation zones as of 1 April 2017. (**a**) is created by d-maps.com (https://d-maps.com/carte.php?num_car=29487, https://d-maps.com/carte.php?num_car=11273). (**b**) is taken from the official website with permission from the administrative officer in Fukushima Prefecture [5].

**Figure 2 ijerph-18-00978-f002:**
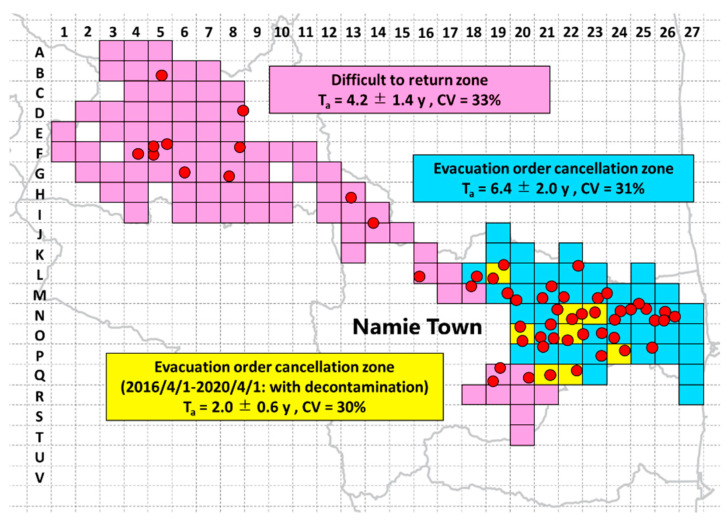
Area classification for which the apparent half-life was calculated, and the location of the monitoring posts. The red circles indicate the location of the monitoring posts used for the analysis, the blue mesh is the difficult-to-return zone, the pink mesh is the evacuation order cancellation zone where the radionuclides decontamination work was carried out before April 2016, and the green mesh is the evacuation order cancellation zone where the radionuclides decontamination work was carried out after April 2016. This map was drawn using a map created by Generic Mapping Tools [35].

**Figure 3 ijerph-18-00978-f003:**
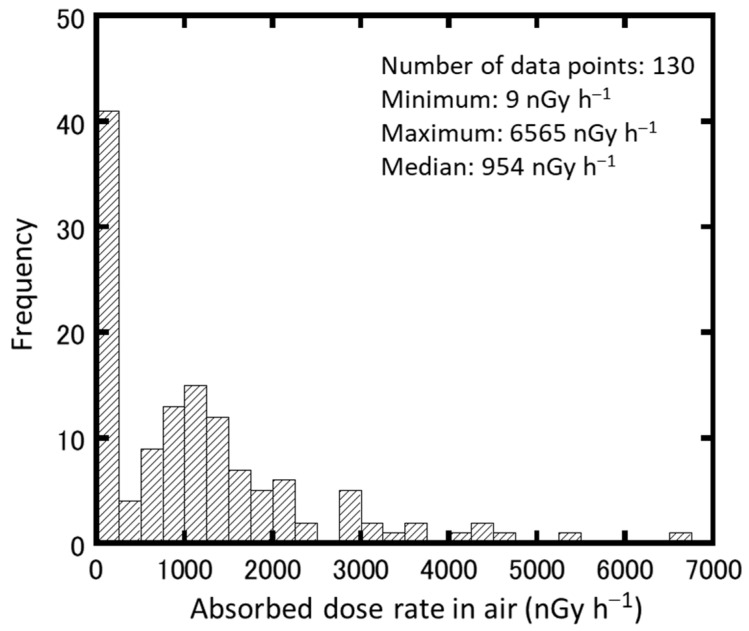
Histogram of absorbed dose rate in air of artificial radionuclides corrected as of 1 April 2020.

**Figure 4 ijerph-18-00978-f004:**
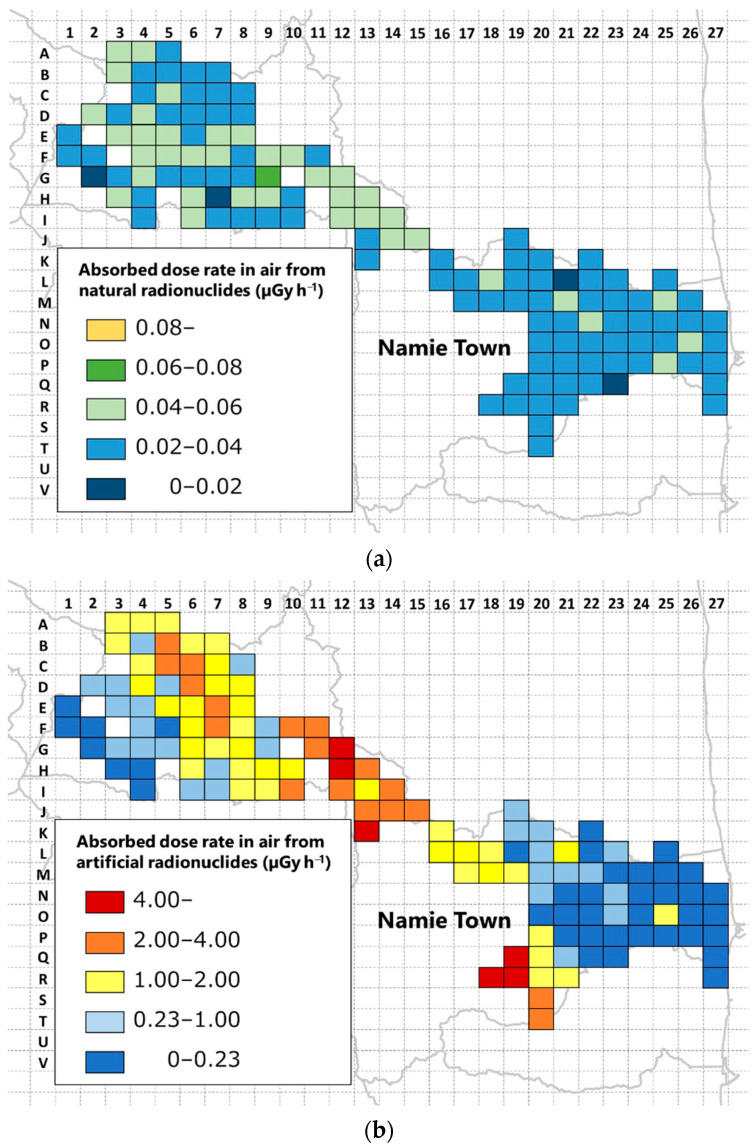
(**a**) Map of absorbed dose rate in air derived from natural radionuclides and (**b**) map of absorbed dose rate in air derived from artificial radionuclides. This map was drawn using a map created by Generic Mapping Tools [35].

**Table 1 ijerph-18-00978-t001:** The estimated value of the quantity of typical radionuclides released into the atmosphere by the Fukushima Daiichi Nuclear Power Plant (FDNPP) accident.

The Estimated Value of the Quantity of Radionuclides Released into the Atmosphere (Bq)							
^132^Te	^131^I	^132^I	^133^I	^133^Xe	^134^Cs	^136^Cs	^137^Cs
2.9 × 10^16^	1.2 × 10^17^	2.9 × 10^16^	9.6 × 10^15^	7.3 × 10^18^	9.0 × 10^15^	1.8 × 10^15^	8.8 × 10^15^

**Table 2 ijerph-18-00978-t002:** Comparison of the half-life of radiocesium in the environment.

Apparent Half-Life of Radiocesium in the Environment (y)				
Evacuation Order Cancellation Zone		Difficult-to-Return Zone	Previously Reported Value [34]	Theoretical Value [34]
Decontaminated before April 2016	Decontaminated on and after April 2016			
6.4	2.0	4.2	3.2	7.8

**Table 3 ijerph-18-00978-t003:** Median (range) estimated absorbed dose rate in air as of 1 April 2020.

	Absorbed Dose Rate in air as of 1 April 2020 (nGy h^−1^)	
	Evacuation Order Cancellation Zone	Difficult-to-Return Zone
Natural radionuclides	28 (25–35)	37 (30–45)
Artificial radionuclides	133 (67–511)	1306 (892–2081)
Total	161 (995–81)	1340 (921–2124)

**Table 4 ijerph-18-00978-t004:** Evaluation of variation of measurements data in a 1 km × 1 km mesh.

Mesh Code	Number of Measurements	Absorbed Dose Rate in Air		
		Average ± Standard Deviation (nGy h^−1^)	Standard Error (nGy h^−1^)	Coefficient of Variation
F5	4	1118 ± 84	42	8%
L22	3	126 ± 33	19	26%
L23	6	312 ± 147	60	47%
M22	5	227 ± 83	37	37%
M24	4	156 ± 14	7	9%
N23	3	147 ± 44	25	30%

**Table 5 ijerph-18-00978-t005:** Estimated annual external exposure effective dose.

	Median (Range) Annual External Exposure Effective Dose (mSv)	
	Evacuation Order Cancellation Zone	Difficult-to-Return Zone
Natural radionuclides	0.19 (0.16–0.23)	0.25 (0.20–0.29)
Artificial radionuclides	0.40 (0.20–1.5)	3.9 (2.7–6.2)
Total	0.55 (0.39–1.7)	4.1 (2.9–6.5)

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
