# Peer review of "Discriminative Measurement of Absorbed Dose Rates in Air from Natural and Artificial Radionuclides in Namie Town, Fukushima Prefecture"

_ijerph, 2021, doi:10.3390/ijerph18030978_

Round 1

Reviewer 1 Report

The article analyses the results of absorbed dose rate measurements from the Fukushima evacuation area. I did not try to reproduce the calculations and statistical test used in the article. However, the chosen methods are valid and scientifically justified.

Obtained results are realistic, logical and applicable for practical purposes. Representation of the results could be improved by e.g. including charts and comparison with other studies. The authors should also clarify some points by adding data on contributing nuclides and elaborating the terminology (especially separating internal and external dose). However, all of these points are minor and can be revised with a reasonable effort. I recommend that the article can be published after minor revision.

My detailed comments are listed below:

Page, line

Comment

 Page 1, line 40

 List of nuclides plausibly contains also e.g. I-129, Xe-133, Xe-135, Kr-85 and Sr-90. I recommend that you add a table including an estimate of total released quantities.

 Page 2, lines 46-47

You could add the annual dose limits for the “emergency evacuation preparation zone” and “warning zone” as well.

 Page 2, section 1

You could separate and quantify the external and internal dose estimates. There may be high local variations, but internal dose in unrestricted use (e.g. farming) would be higher. Therefore (for communication purposes) it is essential to justify why some  areas can be safe for restricted use, but not suitable for agriculture.

 Page 3, section 2.1

What kind of device calibration you used? E.g. a Cs-137 calibration source? This will have an effect of the contribution of other gamma active nuclides with different photon energies. 

 Page 3-4, section 2.2.

Did you perform any spectrometric measurements to separate the contribution from different nuclides? If so, please give an estimate of the quantified result.

Page 4, lines 129-136

I suggest that you compare the measured apparent half-life with an estimate assuming that the only gamma emitter would be e.g. Cs-137.  Please add a chart to visualize the results (to section 3).

Page 4, section 3.1

You mentioned in section 2.2 that some of the data was omitted. Do the reported apparent half-lives contain this data or not? Please quantify how many data point did you use and how many of these may have contained hot spots (significant difference before and after decontamination).

Page 5, figure 2

You could describe how was the evacuation zone initially determined? E.g. is it based on dispersion calculations, dose rate measurements etc.? You don’t have to include it into this article, but I think that you could extend your work by comparing the measured dose rates to natural background in nearby areas around the evacuation zone.

Pages 6 and 7

Please describe which are the main contributing nuclides in natural background radiation in the studied area.

Page 8, lines 283-287

Do you currently have some time limit for the general public? Do you aim to release the area for restricted or unrestricted use after decontamination?

Author Response

We thank the reviewer for your time and effort in evaluating our manuscript.

We have incorporated the suggestions to improve our manuscript.

Please see the attached file for details.

Reviewer 2 Report

The manuscript "Discriminative measurement of absorbed dose rates in air from natural and artificial radionuclides in Namie Town, Fukushima Prefecture" by K. Ogura et al. is focused on the measurements of the gamma-ray pulse-height distribution over the accessible area of Namie Town, Fukushima Prefecture, and on the evaluation of the annual effective dose of external exposure by distinguishing between natural and artificial ingredients. The annual effective dose of external exposure discriminated into natural and artificial ingredients can be utilized for radiation risk communication.

The manuscript contains original results, the overall presentation is convincing and it doesn’t contain errors and miss-formulations.

Only few comments:

  • Authors say that: The gamma-ray pulse-height distributions obtained by the NaI (Tl) scintillation counting system were expanded by a response matrix of 49 rows × 49 columns to provide the absorbed dose rate in air that discriminates between natural and artificial ingredients”.

Please explain this concept more in detail.

  • Please use the term “radionuclides” instead of “ingredients”.
  • Please improve the English language in the paper.

For all these reasons, I suggest to publish the manuscript with minor revision.

Author Response

(The authors gave the same response as above.)
